# Hybridization between Alpine Ibex and Domestic Goat in the Alps: A Sporadic and Localized Phenomenon?

**DOI:** 10.3390/ani12060751

**Published:** 2022-03-17

**Authors:** Barbara Moroni, Alice Brambilla, Luca Rossi, Pier Giuseppe Meneguz, Bruno Bassano, Paolo Tizzani

**Affiliations:** 1Department of Veterinary Sciences, University of Turin, Largo Braccini 2, 10095 Grugliasco, Italy; luca.rossi@unito.it (L.R.); piergiuseppe.meneguz@unito.it (P.G.M.); paolo.tizzani@unito.it (P.T.); 2Alpine Wildlife Research Centre, Gran Paradiso National Park, 10080 Noasca, Italy; alicebrambilla1@gmail.com (A.B.); bruno.bassano@pngp.it (B.B.); 3Department of Evolutionary Biology and Environmental Studies, University of Zurich, 8057 Zurich, Switzerland

**Keywords:** *Capra ibex*, *Capra hircus*, crossbreeding, conservation, livestock, citizen science

## Abstract

**Simple Summary:**

The Alpine ibex (*Capra ibex*) is a protected wild ungulate. The species is known to have a low genetic variability and to occasionally suffer from local population decline as a consequence of epidemic diseases. Another, less investigated, threat for the species’ long-term conservation is represented by the hybridization with feral domestic goats that may breed with ibex if abandoned in the mountain at the end of the summer pasture. Further reproduction and the spread of hybrids may jeopardize the genetic integrity of wild Alpine ibex. By means of an online survey and using a network of experts, we mapped unpublished evidence on observed hybridization events between Alpine ibex and feral domestic goats. The results of this study suggested that hybrids are currently distributed in several countries, and their presence is not a rare event, with some clusters including 4–20 probable hybrids. This calls for more studies clearly quantifying hybrids in Alpine ibex colonies, but also highlights the need for conservation actions aimed at controlling this phenomenon, such as the effective management of domestic herds grazing in Alpine ibex core areas and clear guidelines on hybrid management.

**Abstract:**

The Alpine ibex (*Capra ibex*) is a mountain ungulate living in the European Alps. Although being currently classified as a species of Least Concern (LC) by the IUCN, a potential threat for its long-term conservation is introgression following hybridization with domestic goats (*Capra hircus*). Hybridization has been documented in Switzerland in captive and free ranging animals, although accurate data to assess the extent of this phenomenon in natural conditions in the Alps are lacking. Using an online survey and a network of experts, we collected and mapped unpublished evidence of hybridization events that occurred between Alpine ibex and feral domestic goats from 2000 to 2021. The results of this study showed that hybrids are distributed in most of the Alpine countries, and their presence is not a sporadic event, with some clusters including 4–20 probable hybrids. Our results illustrated the need for establishing a standardized and effective protocol to identify hybrids in the field (such as a formal description of the morphological traits characterizing hybrids), as well as clear guidelines for hybrid management. Even more importantly, this study also highlighted the need for actions aimed at avoiding hybridization, such as the effective management of domestic herds grazing in Alpine ibex core areas.

## 1. Introduction

The rates of hybridization events caused or favoured by anthropogenic activities, such as translocations of organisms and habitat modifications, have increased worldwide in the last decades [1]. Hybridization between wild and domestic species is a special case of anthropogenic hybridization [2,3], and there is large consensus on its potentially harmful effects on the conservation of wild species [4]. Indeed, hybridization with their domestic relatives may jeopardize the long-term genetic conservation of wild species, as shown, for example, in American bison (*Bison bison bison*), where introgression from domestic cattle (*Bos taurus*) has been identified in several bison herds [5]. Hybridization is also described as occurring between wildlife species in other animal classes, such as birds or fishes, where it is a very well-known phenomenon due to human activities, with the introduction of alien species outside their original range [6,7,8]. At the same time, wild–domestic hybridization may pose management and more general conservation issues in the short term, if the introgression leads to changes in the ecology or behavior of the wild species. This is the case, for example, of the Grey wolf (*Canis lupus*), for which hybrids with domestic dogs (*C. lupus familiaris*) are expected to show bolder behavior near humans, with negative consequences for wolf–human relationships [9], or of the wild boar (*Sus scrofa*), for which hybridization with domestic pigs (*S. scrofa domesticus*) may increase the invasiveness potential [10]. The presence of hybrids in the wild may also be an indirect indicator of the health risks for the wild species from the increased risk of pathogen transmission (e.g., African swine fever, brucellosis, pestivirosis) at the livestock/wildlife interface [11]. Finally, the hybridization may cause the local extinction of a species, replaced by hybrids with increased fitness [7]. 

The Alpine ibex (*Capra ibex*) is a mountain ungulate of conservation concern. It is included in the Bern Convention (Convention on the Conservation of European Wildlife and Natural Habitats, Appendix III—Protected Fauna Species, 1979) and in the European Directive 43/92/CEE ‘Habitat’, Annex V (updated with Directive 97/62/CE, 27 October 1997). Currently, the IUCN assessment classifies the Alpine ibex as a species of Least Concern “in view of its wide distribution, presumed large population, and because it is not declining at nearly the rate required to qualify for listing in a threatened category”, but it also declares that “the species needs conservation action to prevent future decline” [12]. The strong bottleneck that occurred in the 19th century [13,14,15] dramatically decreased the genetic variability of the species, whose heterozygosity is one of the lowest registered in wild mammals [3,16], and increased inbreeding and inbreeding depression [17,18]. In addition to the reduced genetic variability, another threat to the species’ long-term conservation is the hybridization with domestic goats. Spatial overlap between livestock and wild herbivores may threaten the long-term conservation of the latter for several reasons: frequent and close contacts between wild and domestic animals may increase resource competition as well as favor cross-transmission of pathogens [11,19,20]. In addition, it may increase the risk of hybridization that, in the case of Alpine ibex, may occur with the domestic goat (*Capra hircus*). In fact, all *Capra* species can interbreed and produce fertile offspring [13]. Hybridization is a serious threat for the Alpine ibex as it may jeopardize its long-term conservation from a genetic perspective [4], eventually resulting in introgression [21]. The discovery of domestic goat alleles in several populations in the Alps, demonstrated that successful hybridization events between Alpine ibex and domestic goats occurred in the past (likely during the 19th century bottleneck), leading to introgression still visible at the Major Histocompatibility Complex (MHC) region [21,22]. Published descriptions of hybrids between Alpine ibex and domestic goats date back to the 1950s, testifying that hybrids could be fertile in both sexes [13,23]. More recently (1991–2001), hybridization has been documented in the Swiss Alps, where a herd of 18 individuals (3 domestic goats and 15 hybrids) has been eradicated by gamekeepers after a ten-year history of poorly controlled livestock dispersion, during the grazing period at the Italian–Swiss border [24]. Hybrids were described as individuals with phenotypic alterations such as increased body size, longer and/or thinner male horns with anomalies in the nodes, and unusual colors on the leg markings, with dark brown spots in young individuals [24]. In addition, reduced reproductive fitness was reported as an effect of hybridization [25] while a potentially improved immune response was proposed as a possible explanation for the long-term introgression observed at the MHC [21].

The hybridization between Alpine ibex and domestic goats may occur in captivity, due to illegal crossbreeding attempts [21,24,25] or in natural conditions when domestic goats are abandoned in mountain pastures at the end of the summer grazing, as the result of poor livestock management. These goats may become feral and join Alpine ibex groups during the reproductive period, in fall and winter [1]. Feral domestic goats can indeed be considered the main cause of these spontaneous hybridization episodes. It is noteworthy that in most of the alpine countries, there are no clear regulations on the management of suspected or confirmed hybrids, and wildlife operators in charge of the management of Alpine ibex colonies are often left alone with the responsibility of taking decisions on an unregulated subject. 

Data on hybrids are mainly available through anecdotal reports, rather than through a systematic approach to the phenomenon. A recent global review [1] listed only six scientific peer-reviewed papers reporting information on hybridization in ibex: two of them were about Iberian ibex [26,27] and four were related to Alpine ibex [21,24,28,29]. In addition, a recent paper [30] mentioning hybridization of Alpine ibex was published after the review from Iacolina et al. [1]. Overall, these papers reported the occurrence of Alpine ibex x domestic goat hybrids in only two countries, only one of which (Switzerland) is part of the Alpine ibex’s natural range. In the other country (Slovakia), hybridization was reported in a population introduced for hunting purposes. In contrast with the limited peer-reviewed information available on Alpine ibex hybridization, relatively abundant grey literature (including ancient books), personal communications, and anecdotal reports pointed towards a non-rare occurrence of hybridization within the Alpine ibex’s natural range. Considering the importance of preserving the genetic integrity of a species for long-term conservation, and the already compromised genetic variability in Alpine ibex, the quantification of the occurrence of this phenomenon is of pivotal importance to carrying out proper preventive and control measures.

The goal of this study was to collect, organize and map unpublished evidence from the field on recent hybridization events between Alpine ibex and domestic goats. To this purpose, a questionnaire was sent to local Alpine wildlife authorities and experts in six countries (Austria, France, Germany, Italy, Slovenia, and Switzerland) to collect relevant information on the presence of suspected or confirmed hybrids, based either on the observation of peculiar phenotypes or on genetic analysis.

## 2. Materials and Methods

### 2.1. Study Area

The study was conducted in the six countries where the Alpine ibex is present in the wild (Austria, France, Germany, Italy, Slovenia, and Switzerland) [31]. An online questionnaire for the collection of data on Alpine ibex hybridization was sent to a list of experts, wildlife managers, institutions, and simply passionate photographers frequenting the Alpine ibex’s natural range.

### 2.2. Data Collection

The survey “Alpine Ibex-Hybridization network” was prepared and distributed using the Google Form tool (Google©) for an online survey. In order to ensure maximum distribution and access to the survey, the content of the form was translated in Italian, English, French and German (Appendix A). The first part of the survey included a presentation of the study and its main objectives, as well as the institution responsible for data collection and details on data processing, in line with European law for data protection (GDPR 2016/679).

The following information was collected:Location of the suspected hybrid (country, administrative division, municipality, up to geographic coordinates if available);Date of the observation;Level of reliability of the observation (confirmed or suspected hybrid);Supporting documentation to validate the observation: written evidence (e.g., reports of institutions, genetic studies), visual evidence (e.g., pictures), no supporting documentation;Details on the suspected hybrid (solitary individuals vs. individuals grouped with Alpine ibex);Information on the concomitant presence of feral domestic goats in the area;Additional comments (free text) from the observer.

The survey was distributed in February 2021 and the deadline for data collection was set for the end of May 2021. In parallel with the distribution of the online survey, we made direct contacts with local experts (by means of emails or phone calls) to ensure the greater accountability of the collected data. 

### 2.3. Data Analysis and Validation

Reports collected through the online survey in the different languages, as well as other data gathered through direct contacts, were grouped in Excel spreadsheets. We obtained three datasets including, respectively: (i) reports of suspected hybrids; (ii) reports of feral domestic goats; (iii) negative reports (i.e., explicit reports of the fact that no suspected hybrids were observed in a given area, to the best of the knowledge of the expert providing the report), with the latter data mainly gathered through direct contacts with local experts. Based on the location of the observation (coordinates or locality/municipality), we assigned each report to the corresponding Alpine ibex colony as defined in [32]. As described in [32], “colony” is used as a working definition to identify the different reintroduced nuclei of Alpine ibex, which are historically known and under different management authorities. We also associated each report of suspected hybrids with the available documentation (e.g., visual documentation, genetic reports). We only included reports relating to the years between 2000 and 2021 in the final dataset. 

All the reports of suspected hybrids associated with visual documentation were independently evaluated by the six authors in order to assess the reliability of the observation (if it was reporting a probable hybrid, an Alpine ibex or a feral domestic goat). The limited information available in the literature [13,23,24], reported that the phenotype of hybrids is different from that of Alpine ibex in terms of alterations in the size and shape of the horns and of the body, and in the color of the coat. However, as no clear and complete description of all the possible hybrid phenotypes exists, each author made their evaluation independently based on his/her knowledge and experience. We then compared the evaluations to assess whether consensus emerged. The reports included in the dataset after the screening were finally classified into three different quality categories (Q1, Q2 and Q3), including, respectively: (Q1) genetically confirmed hybrids; (Q2) suspected hybrids with good quality supporting documentation (visual documentation), and for which an agreement among the authors was reached; and (Q3) suspected hybrids with no supporting documentation or with documentation of low quality, or those for which an agreement among the authors was not reached. As for Q2 and Q3 reports, no genetic confirmation was available, but they were considered reliable, suspected hybrids that after the evaluation fell into Q2 and Q3 categories, were defined as “probable hybrids” (note that, as genetic confirmation was available only for very few cases, we use “probable hybrids” to refer to all the individuals retained in the dataset, also including Q1 reports). When visual documentation was available, we compared the reports (checking the resemblance of the animals, localization, and year of the observation) to avoid multiple counts of the same individual. When no visual documentation was available, multiple Q3 reports from the same locality (e.g., municipality or valley) were not considered as distinct individuals. Likewise, for colonies for which reports of Q1 or Q2 were available, further Q3 reports (with no visual documentation) were not included in the count of probable hybrids. 

The number of reports of probable hybrids and of feral goats retained in the dataset were finally summarized per country. As each hybrid was assigned to the corresponding Alpine ibex colony [32], we then created a map of hybrids and feral goat distribution. We used as a baseline, the map of all the Alpine ibex colonies in the Alps (data originally from Brambilla et al. [32], also included in the IUCN Red List of Threatened Species [12]) and we assigned the number of probable hybrids to the corresponding colony. The reports of feral domestic goats were also associated to the corresponding colony and added to the map. The colonies for which negative reports were available, were classified as colonies with “no hybrids” to differentiate them from those with “no data available” (i.e., no reports collected during the study), and indicated on the map accordingly. For each country, we also calculated the percentage of colonies covered by our study (as the number of colonies for which we received reports of suspected hybrids or negative reports over the total number of colonies for that country), as well as the occurrence of probable hybrids (calculated both as the percentage of the covered colonies of a given country in which probable hybrids were observed, and as the percentage of the total number of colonies of a given country in which probable hybrids were observed). The number of colonies per country was derived from Brambilla et al. [32]. 

## 3. Results

We received N = 43 questionnaires. In addition, we made N = 55 direct contacts with local experts. In total, information and materials were provided by N = 80 different people (as some people provided more than one report). The data gathered during this study covered 39.3% of the Alpine ibex colonies existing in the Alps (N = 70 of 178 colonies).

Before evaluation by the authors, the datasets were composed as follows: (i) the dataset of suspected hybrids contained N = 79 reports; (ii) the dataset of feral domestic goats contained N = 22 reports of feral goats observed in the proximity of Alpine ibex colonies; and (iii) the dataset of negative reports contained N = 49 explicit reports from local experts of the fact that no suspected hybrids were observed in the Alpine ibex colonies. The geographic origin of the reports was unevenly distributed among countries (Table 1), with most of the reports received from Italy. The percentage of the Alpine ibex colonies covered by the survey (i.e., colonies for which either reports of suspected hybrids or negative reports were received) also differed among countries and was not related to the number of reports received (Table 1). 

After the evaluation of all the reports of suspected hybrids, N = 65 reports were retained in the dataset while N = 14 of them were excluded for the following reasons: N = 4 reports were finally classified as domestic goats (one of which was confirmed with genetic analysis); N = 7 reports were classified as Alpine ibex (two of which were confirmed with genetic analysis); and N = 3 reports referred to suspected hybrids that had been raised in captivity.

Following the evaluation and merging of the multiple reports of the same individual, we identified a total of N = 48 probable hybrids (Table 2), observed between 2000 and 2021. For three individuals, genetic confirmation of hybridization was available. The genetic analyses were carried out by specialized laboratories, independently of this study; the methods used to genetically confirm the hybridization involved the genotyping of the suspected individuals at a set of microsatellite or SNP markers and the subsequent calculation of the rate of private loci for the domestic goat (for more details on the microsatellites developed for Alpine ibex genetic monitoring see [16,17,33], while for the SNP-based method see [30]). A further N = 34 individuals were assigned to the category Q2 (i.e., reports of good quality, with reliable visual documentation available and an agreement reached between the authors in the evaluation of the individual as a probable hybrid). Some examples of good quality documentation of the probable hybrids are reported in Figure 1, together with examples of feral domestic goats observed in the proximity of Alpine ibex. The phenotype of the individuals identified as probable hybrids was highly variable and included variations in the size and shape of the horns and of the body outline, and in the coat color compared to Alpine ibex. Some individuals were also hornless (Figure 1). In addition, N = 11 reports were assigned to category Q3 (N = 10 reports coming from reliable local experts, but with no or low-quality visual documentation available, and N = 1 report for which authors disagreed on the possible hybrid phenotype). The number of Q1, Q2 and Q3 reports, as well as the estimated abundance and occurrence of probable hybrids at the country level, is reported in Table 2.

The distribution and abundance of probable hybrids and of feral domestic goats in the Alpine ibex colonies is represented in Figure 2. For most of the colonies, only one probable hybrid was identified, while clusters of more than one individual were observed in five colonies: one in Switzerland, with three individuals, and four in Italy with numbers varying between two and twenty (Figure 2). For other colonies (N = 42), we had explicit reports from local wildlife authorities that no suspected hybrids were observed. Feral domestic goats were reported in 20 out of the 70 investigated colonies. 

## 4. Discussion

This survey represents the first systematic attempt to explore and map the hybridization between Alpine ibex and domestic goats throughout the species’ natural range. The combination of an online survey coupled with the use of an existing network of experts provided valuable results, and the collection of a significant number of reports from most of the Alpine ibex’s geographic range. The results clearly documented that, in the first two decades of the current century, a minimum of 48 probable hybrids were free ranging in Italy, France, Switzerland, and Austria. The occurrence of probable hybrids was highest in Italy, followed by France, Switzerland, and Austria, while no hybrids were signaled in Slovenia and Germany (Table 1, Figure 2). Moreover, evidence of the presence of feral domestic goats in sympatry with Alpine ibex was reported in France, Italy, and Slovenia. In addition, some reports (excluded from the datasets) described hybrids that were born in captivity as a result of crossbreeding between male Alpine ibex and female domestic goats during the grazing season. This phenomenon has already been described in the past [13] as a strategy that was common among goat shepherds, to obtain wilder phenotypic traits and behavior in the domestic breeds.

In most ibex colonies where suspected hybrids were identified, feral goats were also observed in the same or previous years. Although this is not surprising, the opposite did not occur so regularly, suggesting that factors other than the presence of feral goats alone may contribute to the appearance/persistence of hybrids. Despite feral goats being observed throughout the Alps, the results of this survey clearly showed that the presence of hybrids is more frequent in the Western Alps, where the density of Alpine ibex is the highest [32]. Hypothesizing a homogeneous distribution of feral domestic goats, the chances of contacts between them and Alpine ibex increase where Alpine ibex are more abundant. After accounting for the Alpine ibex abundance, however, feral domestic goats, abandoned (either incidentally or voluntarily) at the end of the summer grazing season, appeared as the main background cause of recorded hybridization episodes. It is likely that the presence of feral domestic goats is correlated with the number of domestic goats and goat herds, that pasture in the mountains during summer in a given area. Unfortunately, while official data at the national or infranational scale point towards much greater goat numbers in the Western than in the Eastern Alps, with a ratio well above 2:1, these data were not available at a smaller scale (a more relevant one for the scope of this research), and, therefore, we could not test this hypothesis. 

Regrettably, most Alpine countries lack a clear regulation enabling an effective and timely (e.g., before ibex rut) removal of feral domestic goats, not even when observed in contact with free-ranging Alpine ibex. The same applies for the management of suspected hybrids. We are aware of removal interventions for feral goats and suspected hybrids, carried out under the individual responsibility of park rangers, gamekeepers or hunters, and we have witnessed the discomfort of several in operating in a “grey zone”. 

In five of the twenty-one Alpine ibex colonies where probable hybrids were identified, the presence of more than one hybrid was reported. In particular, a cluster of more than 20 probable hybrids was identified in Italy in the Piedmont region (Figure 2). This cluster was explained by local experts as the abandonment of a whole herd of goats in the late Nineties, following the death of the farmer. More than two decades later, phenotypically unambiguous goats are still present, breed in the area, and are often observed in groups with Alpine ibex. In perspective, the contemporary presence of feral domestic goats, Alpine ibex, and hybrids, with overlapping generations makes it very difficult to identify the hybrids. An extensive genetic investigation of the cluster is, therefore, necessary to understand the level of introgression and of back-crosses. Another cluster of at least four individuals has been observed in Italy, in the Aosta Valley region, a few years after the observation of feral goats in the area. This cluster has been extensively monitored and genetic analysis has confirmed that, at least two of the four individuals observed were back-crosses (e.g., born from the reproduction of an F1 hybrid with a wild Alpine ibex). These observations, together with the case reported by Giacometti [24] in Switzerland, are evidence that, if not managed, the phenomenon of hybridization in Alpine ibex populations can spread and introgression is not unlikely [21].

Based on the scarce scientific literature on this topic [1,21], hybridization between Alpine ibex and domestic goats was expected as occurring sporadically in only two countries. Instead, grey literature and anecdotal reports suggest a non-rare occurrence of the phenomenon. Although this phenomenon and its effect at population level is scarcely known, it is worthy to highlight that several examples of hybridization due to human interventions and activities are well documented in other mammal species, as well as other classes such as birds and fishes, with a wide range of impacts ranging from the local extinction of autochthonous species to the disruption of wildlife population fitness [5,6,7,8]. Our study showed that some hybrids are currently present in most of the European countries. The observation of suspected hybrids in 30% of the colonies investigated (or, more cautiously, in at least 11.8% of all the colonies in the Alps) seems to indicate that the phenomenon is not sporadic. Further studies are, however, needed to assess if, and how, this represents an actual threat to the long-term genetic conservation of the species. The extensive genetic monitoring carried out in the last two decades in a large part of the Alpine ibex range (through random sampling of phenotypically non-suspected individuals) [21,22,30,33] did not find evidence of ongoing hybridization [21]. This could suggest that, if non-adaptive or deleterious [34], domestic goat genes are quickly removed from the Alpine ibex populations (either because of the possibly reduced fecundity of the hybrids [25] or through selection, or dilution, especially if a single hybridization event happens). On the other side, the absence of very recent introgression signals, can also be explained with the hypothesis that hybridization was less common in the past and has increased only recently (for example, because of changes in pasture practices). As no estimate of the number of hybrids in the past was available, it was not possible to draw any conclusion about the trend of the phenomenon or to speculate on possible explanations. This study provides baseline information on the extent of the phenomenon at the present time, that must be monitored in the future to help answer this paramount question for the conservation of the Alpine ibex.

Regardless of temporal trends, however, as the clusters of hybrids identified in this and previous studies [24] clearly showed, hybrids and back-crosses can reproduce, and domestic goat alleles can spread locally in the population, potentially leading to introgression. Indeed, despite the introgression of domestic genes possibly causing outbreeding depression and reduced fecundity or survival [35], it may also result in higher reproductive rates in the introgressed individuals, due to increased heterozygosity (or so-called hybrid vigor [36]). As this has already happened in the past in Alpine ibex [21], current hybridization should be avoided. In order to do so, effective management of domestic herds grazing in Alpine ibex core areas, and proper policies for the management of feral domestic goats and suspected or confirmed hybrids, should be implemented at country and European level. It is important to highlight that the online survey did not allow the report of the absence (to the best of the knowledge of the person who completed it) of hybrids, as the main aim was to focus on observed hybridization events. Since most of the negative reports were gathered at a later time through direct contacts with the experts, the difference between the areas with no data and the areas with negative reports has to be interpreted with caution (Figure 2). The limited information we managed to collect in some countries may also represent a limit of this study. Accordingly, we cannot exclude that hybrids, or feral goats were present in more zones than the ones reported here. Nevertheless, because the Alpine ibex is relatively easily contactable and ibex colonies are often well monitored, e.g., through periodical block counts [32], we believe that our results give a sufficiently reliable overview, at least of the extent of the phenomenon. Moreover, the fact that suspected hybrids were more abundant in the areas with a higher density of Alpine ibex (i.e., the Western Alps) reassures us of the reliability of our survey approach. 

The number of reports was unevenly distributed in space with most of them (either positive or negative) coming from Italy. This, however, does not imply that the data reported are biased. Indeed, the number of reports was not necessarily related to the percentage of colonies for which information was gathered. The different number of observations may in fact be explained by the differences in the organization of Alpine ibex monitoring and management in the different countries (see also [32]). In Switzerland, the management and monitoring of the Alpine ibex is coordinated by the Swiss Ministry for the Environment and by the cantons, that provided accurate aggregated reports (particularly negative reports) allowing us to cover a large part of the Swiss territory, despite the low number of reports. In contrast, no such clear authorities can be identified in the other alpine countries and, to cover the entire territory, we had to contact a large number of local experts. This obligates an alternative strategy, suffering from obvious heterogeneities in the density of contacts and the motivation/ability of the individual expert to extend the requests for information to the relevant people in the field. This strategy worked in Italy but less so in the other countries, Austria, France and Slovenia in particular, where the network of effective contacts proved weaker. Despite the abovementioned limits of the study, our approach of using online surveys, and relying on expert opinions and on data provided by photographers or naturalists, proved to be effective and showed how citizen science can help conservation [37,38], as also demonstrated by recent large-scale successful projects at European scale [39].

A final and very important point to be mentioned, relates to the challenges of identifying the hybrids. The phenotype of the individuals classified as probable hybrids was highly variable. Although some individuals looked more similar to domestic goats, others were phenotypically more similar to Alpine ibex, as the genetic distance between parental species (which may have been pure Alpine ibex in some cases, or first/second generation back-crosses in others) and domestic goats was variable. Compared to Alpine ibex, probable hybrids presented deviations in the coat color, and the size and shape of the body and horns. Some hybrids were hornless. However, it was not possible to define a clear description of a typical hybrid phenotype. This makes identifying hybrids in the field a complex issue. This is confirmed by the fact that 11 reports collected as suspected hybrids were finally classified as domestic goats or Alpine ibex, following our evaluation or genetic analysis. The fact that two phenotypically suspected hybrids were shot in Switzerland around 2010 and that subsequent genetic analysis indicated that they were actually Alpine ibex [30], further highlighted the need for establishing a reliable protocol to identify hybrids and for describing more precisely their phenotypic variation. Genetic tools are already available [30] but, considering costs, time and the bioinformatic background needed, they are not yet applicable for in-field management situations that often need quick decisions to be taken. At the same time, if more samples of suspected hybrids were genetically analyzed, they would firstly, allow determination of hybridization and the rate of introgression at the individual level (necessary for informed management decisions to be taken). Secondly, they would provide new insights on the relationship between the phenotype and hybridization, possibly allowing a formal description of the morphological traits characterizing hybrids, and a standardized protocol to morphologically identify them in the field.

## 5. Conclusions

Our results suggest that the hybridization of Alpine ibex is an underappreciated phenomenon in the Alps, raising important questions on the conservation implications linked to the presence of feral domestic goats, particularly in Alpine ibex core areas. Feral domestic goats in proximity to Alpine ibex represent a threat to the species, both for sanitary [11] and genetic reasons, and this is probably true also for other *Capra* species in other parts of the world [40,41]. Our study reinforces the need and urgence for developing proper policies for future decision-making on the management of hybrids and of feral domestic goats, at country and European levels, and of developing suitable protocols for the identification of hybrids. Further studies should be carried out, focusing on the main questions left unanswered by our work: investigating the colonies for which no data were collected and better quantifying the phenomenon of feral domestic goats, exploring the factors influencing the occurrence and persistence of hybridizations in the Alpine ibex colonies, and investigating the phenotypic variation in hybrids and its relationship with the level of introgression.

## Figures and Tables

**Figure 1 animals-12-00751-f001:**
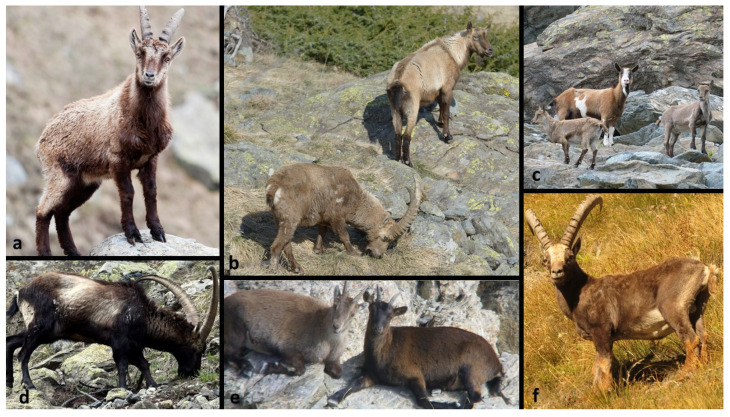
Examples of visual documentation provided as evidence of probable hybridization events (Q2) and of feral goats. (**a**) probable hybrid with deviations of the coat color and abnormal horns and body morphology. Origin: Balme, Piedmont (Italy). Photo Copyright: Simone Miotto. (**b**) probable hybrid (on the right), hornless and with deviations of the coat color, and an Alpine ibex (on the left). Origin: Balme, Piedmont (Italy). Photo Copyright: Luca Giordano. (**c**) female goat in the centre and two hornless young with deviations of the coat color (classified as probable hybrids). Origin: Balme, Piedmont (Italy). Photo Copyright: Luca Giordano. (**d**) probable hybrid with deviations of the coat color. Origin: Balme, Piedmont (Italy). Photo Copyright: Giuseppe Castelli. (**e**) probable hybrid with deviations of the coat color and horns, and abnormal body morphology (on the right), and an Alpine ibex on the left. Origin: Entraunes, Alpi Marittime (France). Photo Copyright: Martin Dhermont. (**f**) probable hybrid with deviations of the coat color. Origin: Bellino, Piedmont (Italy). Photo Copyright: Omar Giordano.

**Figure 2 animals-12-00751-f002:**
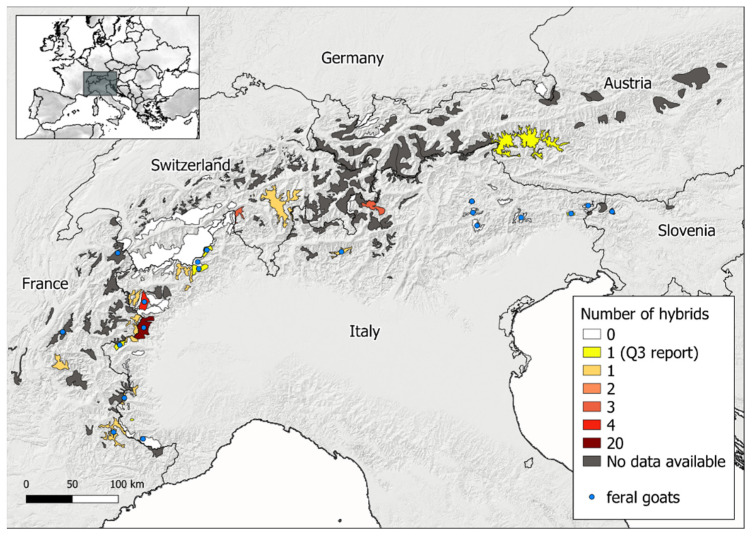
Distribution and abundance of the probable hybrids observed between 2000 and 2021 in the Alps, and the localization of the feral goats observed in the same areas. The map includes the colonies of Alpine ibex (data originally from [32], also included in the IUCN Red List of Threatened Species [12]) colored according to the number of hybrids observed. The orange to dark-red gradient represents colonies with a number of hybrids reported between 1 and 20 (with reports of quality Q1 and Q2). The yellow polygons represent colonies for which a single report of a probable hybrid was received, but the quality of the report was low (Q3: reports coming from reliable local experts, but with no or low-quality visual documentation available). The white polygons represent colonies where no hybrids were reported (negative reports), while the dark grey polygons represent colonies for which no reports were available (neither of the presence nor absence of hybrids). Alpine ibex colonies where feral domestic goats were also observed are represented with a blue dot.

**Table 1 animals-12-00751-t001:** Summary table of the geographic origin of the reports gathered during the study and % of the colonies covered by the study. For each country, the % of colonies covered was calculated as the ratio between the colonies for which either a report of suspected hybrids or a negative report were received and the total number of Alpine ibex colonies of that country. The total number of colonies per country was obtained from [32]. The number of reports does not correspond to the number of colonies covered as some reports covered more than one colony, while other colonies were covered by more than one report from different people.

Country	N Reports Suspected Hybrids	N Reports Feral Goats	N Negative Reports	Total N Reports	Colonies Covered (%)
France	3	3	0	6	16.7% (5/30)
Italy	66	18	40	124	62.7% (42/67)
Switzerland	9	0	4	13	33.3% (15/45)
Germany	0	0	4	4	100.0% (5/5)
Austria	1	0	1	2	7.4% (2/27)
Slovenia	0	1	0	1	25.0% (1/4)
Total	79	22	49	150	39.3% (70/178)

**Table 2 animals-12-00751-t002:** Summary table of the number of Q1, Q2, and Q3 reports, of the total number of probable hybrids and of the estimated abundance and occurrence of probable hybrids at the country level. The number of probable hybrids does not represent the sum of Q1, Q2, and Q3 reports, as multiple Q3 reports from the same locality were not considered as distinct individuals, and Q3 reports were also not included in the count of probable hybrids for the colonies for which Q1 or Q2 reports were available. The occurrence of probable hybrids was calculated both as the percentage of the covered colonies of a given country in which probable hybrids were observed, and as the percentage of the total number of colonies of a given country in which probable hybrids were observed (note that as the % of colonies covered by the study varied greatly between countries, both of these % have to be interpreted with caution). The total number of colonies per country was obtained from [32].

Country	N Q1	N Q2	N Q3	N Probable Hybrids	Hybrid Occurrence (Over Covered Colonies)	Hybrid Occurrence (Over Total Colonies)
France	0	3	0	3	60% (3/5)	10.0% (3/30)
Italy	3	37	7	40	35.7% (15/42)	22.4% (15/67)
Switzerland	0	4	0	4	13.3% (2/15)	4.4% (2/45)
Germany	0	0	0	0	0% (0/5)	0% (0/5)
Austria	0	0	1	1	50% (1/2)	3.7% (1/27)
Slovenia	0	0	0	0	0% (0/4)	0% (0/4)
Total	3	34	11	48	30% (21/70)	11.8% (21/178)

## Data Availability

The data that support the findings of this study are available from the first authors (BM, AB) upon reasonable request.

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
