# Peer review of "Hybridization between Alpine Ibex and Domestic Goat in the Alps: A Sporadic and Localized Phenomenon?"

_animals, 2022, doi:10.3390/ani12060751_

Round 1
Reviewer 1 Report
The manuscript summarized and reviewed the hybridization between Alpine ibex and domestic goat in the 2 Alps as sporadic and localized phenomenon. This manuscript could provide the important clues of the actual distribution of alpine ibex to perform a safety program of this specie.
Therefore, it seems that this paper deserves to be published after adequate responses to the minor comments below.
As regard lines 266 to 270 I suggest reporting some references.
Both in material and methods and results genetic analysis were reported to confirm hybrids. Moreover, in line 293 you reported a reference for Genetic tools as screening of the Capra species. What kind of genetic analysis were performed for this study? Is it the some of the reference of line 293? I think it is appropriate to mention it.
Lines 334-335 report the reference to the figure 2.
Author Response
AUTHORS’ RESPONSE:
Dear Editors and Reviewers,
Thank you for your comments and suggestions, which we found very constructive and helpful to improve the quality of the manuscript. We have now addressed and incorporated relative changes in the manuscript (number of lines refers to the revised manuscript with track changes). In addition to addressing punctual comments by the reviewers, we have edited the summary and the abstract and we have streamlined the methods and the results sections to improve clarity. We have also produced a new map with improved geographic references. In addition, we have largely restructured and expanded the discussion to incorporate the useful suggestions from the reviewers and to better communicate the implications our findings. Finally, we have widened the scientific framework of the study by adding comparisons with other studies in the introduction and in the discussion.
We hope that you will be satisfied with the modifications, and we look forward to hearing from you.
Please find below the detailed point-by-point response to each reviewer, highlighted in red.
The lines in the Authors’ response are referred to the manuscript in track change version.
Reviewer 1
The manuscript summarized and reviewed the hybridization between Alpine ibex and domestic goat in the 2 Alps as sporadic and localized phenomenon. This manuscript could provide the important clues of the actual distribution of alpine ibex to perform a safety program of this specie.
Therefore, it seems that this paper deserves to be published after adequate responses to the minor comments below.
As regard lines 266 to 270 I suggest reporting some references.
Response: In these lines is reported a statement that highlights the results of our study, so we added “Table 1, Figure 2” as reference.
Both in material and methods and results genetic analysis were reported to confirm hybrids. Moreover, in line 293 you reported a reference for Genetic tools as screening of the Capra species. What kind of genetic analysis were performed for this study? Is it the some of the reference of line 293? I think it is appropriate to mention it.
Response: We agree with you that a more detailed description on the genetic analysis used to confirm the hybridization should be provided. We therefore added them in the text (discussion section):
However, we would like to specify that in our study we used genetic reports that have been previously confirmed by dedicated laboratories, thus the genetic analysis should not be considered part of our methodology.
Lines 334-335 report the reference to the figure 2.
Response: Modified as suggested

Reviewer 2 Report
Summary
Thank you for the opportunity to review this manuscript for Animals. This study uses survey and interview data to document and represent the phenomenon of hybridisation between the Alpine ibex and the domestic goat. The novel aspect of the paper is that the investigation is carried out over the entire natural distribution of the Alpine ibex, spreading over 6 countries, with data collected in the same manner. This paper reduces the knowledge gap about hybridisation between wild and domestic species, which is likely to increase due to anthropisation. Through its results, this article can justify the development of future research projects aimed at limiting this phenomenon, notably by improving the identification of hybrids.
The question is relevant, and the methods are appropriate. However, the manuscript lacks precision and clarity on the methods and results, and there are discrepancies between the results and the discussion. The text should be clarified in several aspects, and the discussion can be strengthened by limiting repetition with what is said in the results.
General comments
I feel that the simple summary is a bit long and like what is said in the abstract; some sentences are almost identical. The simple summary should describe your study simply and concisely for a wide audience. Rephrasing and reducing the simple summary may make it more understandable (see my comments below).
In general, you use a lot of passive voice, but you could use more active voice in the methods and results parts, which would make them clearer and more easily understandable, in my view. Also, the analyses and results are yours, so the active voice would be more appropriate.
Be consistent in the terms you use, e.g. "hybridization" which you were using throughout becomes "hybridization" line 42; "suspected hybrids" are also called "probable hybrids" or "potential hybrids", which is confusing (unless you don't mean the same thing, in which case it would be useful to explain); "Domestic goat” is sometimes called feral or abandoned goat; the feral goat being the domestic goat when it has established itself in the wild, you could just use the term domestic goat.
There is a general ambiguity in the methodological details and results (e.g., in the definition of negative reports). In general, the results could be presented differently to facilitate their understanding. The lack of definition of certain terms, such as a colony, is an issue.
Specific comments
Simple summary
Lines 12-15: I suggest a more concise formulation. I am not sure that mentioning the threat of disease outbreaks to the conservation of the species is useful here. A more concise sentence can be formulated here, e.g. “The Alpine ibex (Capra ibex) is a protected wild ungulate but under long-term threat due to possible hybridisation with domestic goats, sometimes left in the mountains at the end of the summer grazing season. Breeding between these species may alter the genetic diversity of the Alpine ibex and lead to its extinction.”
Lines 16-20: these two sentences [“The main goal” ... “from 2000 and 2021”] say something similar. What do you think about rewording them and shortening them to be more concise? e.g., "Using an online survey and a network of experts, we mapped published and unpublished evidence on observed hybridization events between the Alpine ibex and the domestic goat."
Line 17: What do you mean by "distribution in time"? With such a term, I would expect you to explore differences in the occurrence of these events between periods of time (e.g., months or years), which is not the case here. I thought you might be saying that because you have restricted the observation period to 2000-2021, but if that is the case, then I don't think you are looking at the distribution over time of these events, as there is no comparison with earlier periods (except for a few words in the discussion, but that is not your results).
Line 20: not sure if mentioning the observation period (between 2000 and 2021) is necessary for the simple summary.
Line 21: How can you say that the events are "more common than expected"? You have not done any analysis to be able to say this; you only state what has been found in the previous literature in the discussion section, but, in my view, this does not constitute proof (same comment in the abstract).
Lines 22-25: A simpler sentence (and different from what is said in the abstract) can be formulated, e.g., “Our results suggest that domestic goat herds grazing in the core areas of the Alpine ibex should be better monitored (one example of how might be useful?). Investigations on morphological traits should be deepened when hybridization is suspected in order to be able to properly regulate this problem”.
Line 24: you should add an 'and' between 'core areas' and 'monitoring surveys'.
Abstract
Line 28: an 'it' is missing? => "Although it is currently".
Line 42: you write "hybridisation” when in the rest of the manuscript you write “hybridization”.
Introduction
Line 58: An example of the consequences of hybridization would be useful.
Lines 64-66: again, I do not see the point of mentioning the possible transmission of pathogens between species. While this process may indeed threaten the survival of the Alpine ibex, the use of the same space (without being in contact) may also threaten the ibex, e.g., because of competition for resources. You could rather start this paragraph by saying why C. ibex and C. hircus can be in contact (same diet, same use of space), which favors reproduction and thus hybridization between these 2 species.
Line 72: “description” instead of “ddescription”.
Line 95: “The goal” instead of “The main goal”.
Lines 98-99: you say five countries, but you list six.
Materials and Methods
Line 110: “we had direct contacts” instead of “the authors made”.
Line 132: couldn't "abandoned/feral goat" be replaced by "domestic goat"?
Section 2.3 reads quite well, but on closer inspection, it is not entirely clear how the reports were selected, sorted, and validated. I suggest reorganizing the section a bit and being more precise and consistent in the terms used (see my comments below).
Line 137: You might want to start this section with a sentence in active voice rather than passive. In general, this whole section could be clearer if you use more active voice, especially as it is your analysis.
Lines 140-141: What do you mean by “only”? Is it only reports of suspected hybrids that are included in the final dataset (i.e., not negative reports) or does it mean that you had reports of suspected hybrids from before 2000 but decided not to include them? If the latter, what about the other reports (i.e., negative or with goats)? For the latter, you do not indicate anywhere, unless I am mistaken, that they are in the period 2000-2021.
Line 143: "potential hybrid" is named "possible hybrid" line 147 and "suspected hybrids" line 151. I have the impression that these 3 terms mean the same thing, if so, please use one term throughout the manuscript. If not, please clearly explain the difference between these terms.
Line 147: "the authors made their evaluation" can be replaced by "we evaluated", easier to read and understand.
Line 148: "the author's evaluations" can be replaced by "our evaluations".
Line 160: for those unfamiliar with the species, it is important that you define the term “colony”. Is it a group or a population? In line 164 you seem to say that colonies are defined according to the definition given in reference 21, but on the one hand, this comes after you have already mentioned the existence of colonies (line 160), and on the other hand referring to an article does not give any information. I think it is necessary to explain what a colony is, either in the introduction or in the methods.
Line 160: How did you classify the reports in the colonies? You have classified the reports obtained in different quality categories and then you mention that the colonies could have reports of Q1, Q2 and Q3. I assume that the location of the colonies is known (but how?) and that by comparing it with the location given in the reports you were able to assign the reports to the colonies, but it might be useful to say so. This may be what you say below, line 164, but it needs to be explained earlier; otherwise, the reader cannot properly understand this sentence.
Lines 162-166: should be a new paragraph, as this is about analysis, not dataset preparation. Therefore, the sentence after ("The colonies from which ..." lines 166-170) should not be here, and possibly earlier in the methods.
Line 167: "were questioned and" can be deleted.
Line 167-169: the definition of negative reports does not fully correspond with the one given below in the results. Here you say it is when no suspected hybrids were observed, whereas in the results (line 178) you also add that it is when no goats were observed.
Lines 167-170: Perhaps it would help to state at the outset the different reports you obtained with a clear and precise definition of each one (i.e., negative reports = no hybrids and no goats, positive reports = hybrids and/or goats)? I found that learning about the existence of negative reports came a little late in the section. After defining the reports, you could then clearly state that all reports are for the period 2000-2021 if this is the case, and then focus only on the reports where hybrids were suspected, and thus how you evaluate them.
Results
It is a bit difficult to keep track of the number of reports in each category. Wouldn't a schematic tree showing the classification steps and the number of reports per report category and quality category make the results clearer? Perhaps the explanations for the exclusion of some reports could be included. This would make the text smaller and easier to understand. This could also make the steps described in 2.3 clearer.
Line 172: You might want to start this section with a sentence in active voice rather than passive: "We received 43 questionnaires and had 55 direct contacts ...".
Lines 175-179: This paragraph seems to be a repetition of what is said in Table 1 (in the totals). It is therefore not necessary to mention it in text form.
Line 178: The definition of what is a negative report differs from the one given in methods (see comment above).
Line 182: positive reports. This is the first time you mention it.
Lines 191-194: can be reworded more clearly, e.g. "We excluded 14 reports of suspected hybrids from the dataset for the following reasons: 4 reports were finally classified as domestic goats (one confirmed with genetic analysis and three after our convergent evaluations), ...". Furthermore, in the discussion, you give further explanations as to why the reports were excluded (lines 286-289); these explanations should come here, not in the discussion.
Line 194: Could you also indicate the total number of reports on which your results are based after this exclusion?
Line 195: A simpler sentence can be formulated: "Following our evaluations and the merging of multiple reports of the same individual, we identified ...".
Line 209: Since here you show the results per country, I would start a new paragraph. Rather than explaining what Figure 2 represents, more precise results can be formulated here, e.g., the number and abundance of probable hybrids were heterogeneous between countries, with more hybrids in Italy.
Lines 213-217: the number of colonies studied (total or per country) is not very clear. Is it 64 (line 214) or 70 (line 217)? The presence of goats was noted in 20 colonies and for 42 other colonies, no goats were identified, what about the 2 other colonies (as you mention 64 colonies)? Although you give the percentage of colonies surveyed, it might be useful to also know the number of colonies known per country and the number you surveyed. This could be given in a new table. You could also give the number of colonies with suspected hybrids and/or with goats.
Table 1: 1) I would suggest that the organization and wording of the table be changed slightly so that the reader understands that the "N reports" column is the total of the "N reports hybrids" / "N reports goats" and "N negative reports" columns. 2) I would not indicate the number of probable hybrids per country in this table for 2 reasons: (i) this is already mentioned in the text (lines 210-212); (ii) since the number of reports as well as the % of colonies studied given here are before evaluation of the reports, it is not coherent, in my view, to indicate in this table the probable number of hybrids, since this result is obtained after evaluation of the reports.
Discussion
Lines 241-257: This whole section is about the context of your study, i.e., what is already available in the literature and what is missing. In this sense, this part should be in the introduction to justify your study. The first paragraph of the discussion usually focuses on a summary of your aim and results, although writing 1-2 short sentences on the reasons for your study is possible.
Line 246: Reference 26 is on which species?
Line 257: “Thus” is not necessary.
Line 263: “From a geographic perspective” can be deleted.
Lines 265-267: These are new results. The numbers should be given in the Results section and not in the Discussion. This could be in a new table showing the results of the presence/absence of hybrids and/or goats per colony and per country (see my comment for lines 213-217).
Lines 268-271: These are also new results, which were not included in the Results section. The number of reports with clusters and the number of clusters per country should be given in the results.
Line 279: "after our evaluation" instead of “after evaluation by the authors”.
Lines 282-284: I appreciate this paragraph about the difficulty of clearly identifying hybrids between the 2 species, but the known differences between the hybrids and the two species have already been said in the introduction, in the methods and results. I think it is therefore not necessary to repeat it here, especially as this discussion can be perfectly understood without explaining again the differences between Alpine ibex and hybrids.
Lines 286-289: you repeat what was said in the results. Say it but with less detail, e.g., "This is confirmed by the fact that 11 reports initially identifying suspected hybrids were finally classified as domestic goats or Alpine ibexes, following our evaluation or genetic analysis.”
Line 291: Is it necessary to state that it is figs. 6b and 6c and the second and third individuals?
Line 301-302: Could be rephrased "the results of this survey clearly showed that the presence of hybrids is more frequent in the Western Alps, where the density of Alpine ibex is the highest [21]”.
Lines 334-335: Indicating the colour of the areas on the map is not relevant to the discussion
Line 336: “cannot” instead of “can't”
Line 341: "as previously stated" can be removed.
Line 343: "As already stated before" is not necessary.
Line 352: "presented in Table 1" should be removed. This does not have to be in the discussion.
Line 357: You can delete "It is therefore not surprising that".
Conclusion
Line 361: no need for the word “in conclusion”
Line 362: What are the health implications of hybridisation? It might be useful to give an example, perhaps even in the introduction.
Figure 2.
The figure printed in black and white is unreadable. This can be improved by using more readable characters (eg, hatches) and colours.
Author Response
Reviewer 4
Thank you for the opportunity to review this manuscript for Animals. This study uses survey and interview data to document and represent the phenomenon of hybridisation between the Alpine ibex and the domestic goat. The novel aspect of the paper is that the investigation is carried out over the entire natural distribution of the Alpine ibex, spreading over 6 countries, with data collected in the same manner. This paper reduces the knowledge gap about hybridisation between wild and domestic species, which is likely to increase due to anthropisation. Through its results, this article can justify the development of future research projects aimed at limiting this phenomenon, notably by improving the identification of hybrids.
The question is relevant, and the methods are appropriate. However, the manuscript lacks precision and clarity on the methods and results, and there are discrepancies between the results and the discussion. The text should be clarified in several aspects, and the discussion can be strengthened by limiting repetition with what is said in the results.
Response: thank you very much for your helpful comments and suggestions, we have edited the text accordingly and you can find our replies in red, below each comment.
General comments
I feel that the simple summary is a bit long and like what is said in the abstract; some sentences are almost identical. The simple summary should describe your study simply and concisely for a wide audience. Rephrasing and reducing the simple summary may make it more understandable (see my comments below).
Response: Abstract and simple summary have been edited following your specific comments.
In general, you use a lot of passive voice, but you could use more active voice in the methods and results parts, which would make them clearer and more easily understandable, in my view. Also, the analyses and results are yours, so the active voice would be more appropriate.
Response: thank you for pointing this out. We have reduced the use of passive voice where appropriate.
Be consistent in the terms you use, e.g. "hybridization" which you were using throughout becomes "hybridization" line 42; "suspected hybrids" are also called "probable hybrids" or "potential hybrids", which is confusing (unless you don't mean the same thing, in which case it would be useful to explain); "Domestic goat” is sometimes called feral or abandoned goat; the feral goat being the domestic goat when it has established itself in the wild, you could just use the term domestic goat.
Response: we have now clearly defined in the text the different terms (with “suspected hybrids” referring to the reports before the evaluations and “probable hybrids” referring to the reports retained in the dataset after the evaluation. As genetic confirmation was not available for most of them, we have decided to use “probable hybrids” for all of them. Despite we realize that this makes some sentence a bit “heavy”, we also didn’t want to convey too much certainty on the probable hybrids that were not genetically confirmed by just using “hybrids” or “confirmed hybrids” for all of them. (at the same time, we believe that repeating every time “confirmed or probable hybrids” would have been really too much).
There is a general ambiguity in the methodological details and results (e.g., in the definition of negative reports). In general, the results could be presented differently to facilitate their understanding. The lack of definition of certain terms, such as a colony, is an issue.
Response: we have better defined the colonies and negative reports and made more clear how we processed and analyzed the data.
Specific comments
Simple summary
Lines 12-15: I suggest a more concise formulation. I am not sure that mentioning the threat of disease outbreaks to the conservation of the species is useful here. A more concise sentence can be formulated here, e.g. “The Alpine ibex (Capra ibex) is a protected wild ungulate but under long-term threat due to possible hybridisation with domestic goats, sometimes left in the mountains at the end of the summer grazing season. Breeding between these species may alter the genetic diversity of the Alpine ibex and lead to its extinction.”
Response: As suggested, we edited the sentence. However, we preferred to keep a reference to the diseases as we think they should be taken into account when referring to general threats for the conservation of the species: “”.
Lines 16-20: these two sentences [“The main goal” ... “from 2000 and 2021”] say something similar. What do you think about rewording them and shortening them to be more concise? e.g., "Using an online survey and a network of experts, we mapped published and unpublished evidence on observed hybridization events between the Alpine ibex and the domestic goat."
Response: We have modified the sentence as suggested
Line 17: What do you mean by "distribution in time"? With such a term, I would expect you to explore differences in the occurrence of these events between periods of time (e.g., months or years), which is not the case here. I thought you might be saying that because you have restricted the observation period to 2000-2021, but if that is the case, then I don't think you are looking at the distribution over time of these events, as there is no comparison with earlier periods (except for a few words in the discussion, but that is not your results).
Response: we have deleted the sentence as we had no data to explore the development in time of the phenomenon.
Line 20: not sure if mentioning the observation period (between 2000 and 2021) is necessary for the simple summary.
Response: we have deleted the observation period.
Line 21: How can you say that the events are "more common than expected"? You have not done any analysis to be able to say this; you only state what has been found in the previous literature in the discussion section, but, in my view, this does not constitute proof (same comment in the abstract).
Response: this point was raised by reviewer 3 as well. With “hybrids more common than expected” we meant that, according to the sparse scientific literature on this topic, hybrids of Alpine ibex x domestic goat were only a limited, sporadic phenomenon. Our study (as well as the anecdotical information referred by experts in the field) proved this hypothesis wrong, as not only many phenotypic hybrids have been reported in the Alps, but also in our results there might be an underestimation, as stated in Discussion in line 363 “Accordingly, we can’t exclude that hybrids or feral goat were present in more zones than the ones reported here.”
Nonetheless, to clarify the concept, we modified the sentence
Lines 22-25: A simpler sentence (and different from what is said in the abstract) can be formulated, e.g., “Our results suggest that domestic goat herds grazing in the core areas of the Alpine ibex should be better monitored (one example of how might be useful?). Investigations on morphological traits should be deepened when hybridization is suspected in order to be able to properly regulate this problem”.
Response: we modified the sentenceto make it more clear
Line 24: you should add an 'and' between 'core areas' and 'monitoring surveys'.
Response: the sentence has been rephrased as mentioned above.
Abstract
Line 28: an 'it' is missing? => "Although it is currently".
Response: sentence modified in “Although being currently classified”
Line 42: you write "hybridisation” when in the rest of the manuscript you write “hybridization”.
Response: we have modified as suggested
Introduction
Line 58: An example of the consequences of hybridization would be useful.
Response: several examples of the consequences of hybridization have been reported in the introduction
Lines 64-66: again, I do not see the point of mentioning the possible transmission of pathogens between species. While this process may indeed threaten the survival of the Alpine ibex, the use of the same space (without being in contact) may also threaten the ibex, e.g., because of competition for resources. You could rather start this paragraph by saying why C. ibex and C. hircus can be in contact (same diet, same use of space), which favors reproduction and thus hybridization between these 2 species.
Response: we have rephrased the sentence in order to add all the possible threat coming from spatial overlap between wild and domestic species.
Line 72: “description” instead of “ddescription”.
Response: modified as suggested
Line 95: “The goal” instead of “The main goal”.
Response: modified as suggested
Lines 98-99: you say five countries, but you list six.
Response: corrected in six countries as suggested
Materials and Methods
Line 110: “we had direct contacts” instead of “the authors made”.
Response: edited as suggested.
Line 132: couldn't "abandoned/feral goat" be replaced by "domestic goat"?
Response: we changed it to feral domestic goat. Throughout the text, when referring to feral goats we have decided to use the term feral domestic goat (with domestic goat being the common name of the species C. hircus)
Section 2.3 reads quite well, but on closer inspection, it is not entirely clear how the reports were selected, sorted, and validated. I suggest reorganizing the section a bit and being more precise and consistent in the terms used (see my comments below).
Response: we have largely restructured the section to improve readability and clarify some of the below mentioned issues. We first added a clearer description of the data collected and then separated the different elaboration done for each kind of data. Please, see the new version of the section 2.3
Line 137: You might want to start this section with a sentence in active voice rather than passive. In general, this whole section could be clearer if you use more active voice, especially as it is your analysis.
Response: When possible, we have switched to active voice.
Lines 140-141: What do you mean by “only”? Is it only reports of suspected hybrids that are included in the final dataset (i.e., not negative reports) or does it mean that you had reports of suspected hybrids from before 2000 but decided not to include them? If the latter, what about the other reports (i.e., negative or with goats)? For the latter, you do not indicate anywhere, unless I am mistaken, that they are in the period 2000-2021.
Response: following the restructuring of the paragraph, this sentence was deleted.
Line 143: "potential hybrid" is named "possible hybrid" line 147 and "suspected hybrids" line 151. I have the impression that these 3 terms mean the same thing, if so, please use one term throughout the manuscript. If not, please clearly explain the difference between these terms.
Response: we acknowledge that there was a lack of homogeneity in the terms used. We have changed the text to avoid this. We used “suspected hybrid” when referring to the report before our evaluation, and “probable hybrid” or “hybrid” when referring to the evaluated report. As no genetic confirmation was available, but the reports were considered reliable, suspected hybrids whose reports fell into Q2 and Q3 categories were defined as “probable hybrids”. However, as genetic confirmation was available only for very few cases, we interchangeably referred as hybrids or probable hybrids when referring to all the reports (to both Q1 and Q2 and Q3).
Line 147: "the authors made their evaluation" can be replaced by "we evaluated", easier to read and understand.
Line 148: "the author's evaluations" can be replaced by "our evaluations".
Response: we have changed the passive voice to active in several sentences, but we preferred to leave these two sentences as they were.
Line 160: for those unfamiliar with the species, it is important that you define the term “colony”. Is it a group or a population? In line 164 you seem to say that colonies are defined according to the definition given in reference 21, but on the one hand, this comes after you have already mentioned the existence of colonies (line 160), and on the other hand referring to an article does not give any information. I think it is necessary to explain what a colony is, either in the introduction or in the methods.
Response: we have added in the text the definition of Alpine ibex colony at the beginning of 2.3 paragraph in the methods section.
Line 160: How did you classify the reports in the colonies? You have classified the reports obtained in different quality categories and then you mention that the colonies could have reports of Q1, Q2 and Q3. I assume that the location of the colonies is known (but how?) and that by comparing it with the location given in the reports you were able to assign the reports to the colonies, but it might be useful to say so. This may be what you say below, line 164, but it needs to be explained earlier; otherwise, the reader cannot properly understand this sentence.
Response: as we added the definition of Alpine ibex colony earlier in the text, at the beginning of 2.3 paragraph in the methods section and we believe that this is now clearer
Lines 162-166: should be a new paragraph, as this is about analysis, not dataset preparation. Therefore, the sentence after ("The colonies from which ..." lines 166-170) should not be here, and possibly earlier in the methods.
Response: We have restructured the paragraph and better separated the different stages of data preparation and analysis.
Line 167: "were questioned and" can be deleted.
Response: deleted as suggested.
Line 167-169: the definition of negative reports does not fully correspond with the one given below in the results. Here you say it is when no suspected hybrids were observed, whereas in the results (line 178) you also add that it is when no goats were observed.
Response: we realize that the sentence in the earlier version of the manuscript was not clear, and we therefore modified it. Negative reports referred to reports that no suspected hybrids were observed in the area.
Lines 167-170: Perhaps it would help to state at the outset the different reports you obtained with a clear and precise definition of each one (i.e., negative reports = no hybrids and no goats, positive reports = hybrids and/or goats)? I found that learning about the existence of negative reports came a little late in the section. After defining the reports, you could then clearly state that all reports are for the period 2000-2021 if this is the case, and then focus only on the reports where hybrids were suspected, and thus how you evaluate them.
Response: we have restructured the whole paragraph. At the beginning we described all the data available (including negative reports that were clearly defined) so we hope that the text has now improved in clarity.
Results
It is a bit difficult to keep track of the number of reports in each category. Wouldn't a schematic tree showing the classification steps and the number of reports per report category and quality category make the results clearer? Perhaps the explanations for the exclusion of some reports could be included. This would make the text smaller and easier to understand. This could also make the steps described in 2.3 clearer.
Response: following restructuring of data analysis paragraph, we have also changed the structure of the results also changing the table. We now hope that the results are easier to follow.
Line 172: You might want to start this section with a sentence in active voice rather than passive: "We received 43 questionnaires and had 55 direct contacts ...".
Response: we have reduced the use of passive voices as suggested.
Lines 175-179: This paragraph seems to be a repetition of what is said in Table 1 (in the totals). It is therefore not necessary to mention it in text form.
Response: we have modified the sentence in order to link it more clearly to paragraph 2.3 and therefore we have decided to keep the numbers in the text as well. We believe that this sentence also helps to better understand what is reported in table 1.
Line 178: The definition of what is a negative report differs from the one given in methods (see comment above).
Response: we have better explained this issue in the text. Negative reports were indeed those reporting the absence of suspected hybrids only.
Line 182: positive reports. This is the first time you mention it.
Response: positive reports was referring to reports of suspected hybrids, but we realized that this was not clear. We have therefore changed the words to make it explicit.
Lines 191-194: can be reworded more clearly, e.g. "We excluded 14 reports of suspected hybrids from the dataset for the following reasons: 4 reports were finally classified as domestic goats (one confirmed with genetic analysis and three after our convergent evaluations), ...". Furthermore, in the discussion, you give further explanations as to why the reports were excluded (lines 286-289); these explanations should come here, not in the discussion.
Response: we have reworded the sentence to improve readability. The sentence in the discussion was also modified accordingly.
Line 194: Could you also indicate the total number of reports on which your results are based after this exclusion?
Response: we have added in the text the number of reports retained in the dataset (65).
Line 195: A simpler sentence can be formulated: "Following our evaluations and the merging of multiple reports of the same individual, we identified ...".
Response: we have modified the text as suggested.
Line 209: Since here you show the results per country, I would start a new paragraph. Rather than explaining what Figure 2 represents, more precise results can be formulated here, e.g., the number and abundance of probable hybrids were heterogeneous between countries, with more hybrids in Italy.
Response: a new paragraph has been added as suggested.
Lines 213-217: the number of colonies studied (total or per country) is not very clear. Is it 64 (line 214) or 70 (line 217)? The presence of goats was noted in 20 colonies and for 42 other colonies, no goats were identified, what about the 2 other colonies (as you mention 64 colonies)? Although you give the percentage of colonies surveyed, it might be useful to also know the number of colonies known per country and the number you surveyed. This could be given in a new table. You could also give the number of colonies with suspected hybrids and/or with goats.
Response: the correct number of colonies covered by the study was 70 (64 referred to an older version of the results, before adding the last reports, thanks for pointing this out and we are sorry for having overseen this). In addition, the negative reports were only related to the absence of hybrids. We didn’t analyze negative reports of feral goats, and this was clarified in the text. Finally, we also added the total number of colonies per country (as reported in ref. Brambilla et al., 2020) in Table 1.
Table 1: 1) I would suggest that the organization and wording of the table be changed slightly so that the reader understands that the "N reports" column is the total of the "N reports hybrids" / "N reports goats" and "N negative reports" columns. 2) I would not indicate the number of probable hybrids per country in this table for 2 reasons: (i) this is already mentioned in the text (lines 210-212); (ii) since the number of reports as well as the % of colonies studied given here are before evaluation of the reports, it is not coherent, in my view, to indicate in this table the probable number of hybrids, since this result is obtained after evaluation of the reports.
Response: We have modified the table following your suggestion.
Discussion
Lines 241-257: This whole section is about the context of your study, i.e., what is already available in the literature and what is missing. In this sense, this part should be in the introduction to justify your study. The first paragraph of the discussion usually focuses on a summary of your aim and results, although writing 1-2 short sentences on the reasons for your study is possible.
Response: as also requested by Reviewer 3, we moved the entire paragraph in Introduction section and slightly modified it.
Line 246: Reference 26 is on which species?
Response: the reference reports hybridization episodes of Alpine ibex. The sentence has been rephrased as follows: “In addition, a recent paper [23] mentioning hybridization of Alpine ibex was published after the review from Iacolina et al. [9].”
Line 257: “Thus” is not necessary.
Response: deleted as suggested.
Line 263: “From a geographic perspective” can be deleted.
Response: modified as suggested
Lines 265-267: These are new results. The numbers should be given in the Results section and not in the Discussion. This could be in a new table showing the results of the presence/absence of hybrids and/or goats per colony and per country (see my comment for lines 213-217).
Lines 268-271: These are also new results, which were not included in the Results section. The number of reports with clusters and the number of clusters per country should be given in the results.
Response: we have moved this numbers to the results section
Line 279: "after our evaluation" instead of “after evaluation by the authors”.
Response: modified as suggested.
Lines 282-284: I appreciate this paragraph about the difficulty of clearly identifying hybrids between the 2 species, but the known differences between the hybrids and the two species have already been said in the introduction, in the methods and results. I think it is therefore not necessary to repeat it here, especially as this discussion can be perfectly understood without explaining again the differences between Alpine ibex and hybrids.
Response: We agree that there might be some repetitions on this, but as you and other reviewers mentioned, the phenotypic details of hybrids are the most debated and perhaps ambiguous characteristics. Thus, we believe that keeping this kind of information in the discussion section might be useful for the reader. However, as the discussion has been edited according to your and reviewer n 3 suggestions, this part of the discussion is now towards the end and it has been expanded in order to highlight the difficulties of identifying hybrids and the need of further studies aiming at establishing a shared and effective protocol for hybrid identification.
Lines 286-289: you repeat what was said in the results. Say it but with less detail, e.g., "This is confirmed by the fact that 11 reports initially identifying suspected hybrids were finally classified as domestic goats or Alpine ibexes, following our evaluation or genetic analysis.”
Response: modified as suggested.
Line 291: Is it necessary to state that it is figs. 6b and 6c and the second and third individuals?
Response: the words in brackets have been deleted as suggested.
Line 301-302: Could be rephrased "the results of this survey clearly showed that the presence of hybrids is more frequent in the Western Alps, where the density of Alpine ibex is the highest [21]”.
Response: modified as suggested.
Lines 334-335: Indicating the colour of the areas on the map is not relevant to the discussion
Response: we deleted the color indications in brackets
Line 336: “cannot” instead of “can't”
Response: modified as suggested.
Line 341: "as previously stated" can be removed.
Response: the words have been deleted as suggested.
Line 343: "As already stated before" is not necessary.
Response: the words have been deleted as suggested.
Line 352: "presented in Table 1" should be removed. This does not have to be in the discussion.
Response: the words have been deleted as suggested.
Line 357: You can delete "It is therefore not surprising that".
Response: the words have been deleted as suggested.
Conclusion
Line 361: no need for the word “in conclusion”
Response: the words have been deleted as suggested.
Line 362: What are the health implications of hybridisation? It might be useful to give an example, perhaps even in the introduction.
Response: examples have been provided in the introduction
Figure 2.
The figure printed in black and white is unreadable. This can be improved by using more readable characters (eg, hatches) and colours.
Response: we have edited the figure according to reviewer 2 suggestion. We have also tried to use different patterns/colors as you suggested but given the small size of some of the colonies, we weren’t able to find a satisfying alternative combination of colors/patterns. As the journal is only published online, hence with figures in colors, we hope that this won’t be too much of an issue.

Reviewer 3 Report
Comments to the authors
In this paper, authors investigated through an online survey and a literature review, the occurrence of Alpine ibex x Capra hircus hybrids in the Alps. Investigating almost half of the total number of colonies known in the species range, they report the presence of 48 suspected hybrids. They conclude that their study reveals a higher than expected number of hybrids and highlight the need of a better management policy of hybrids and feral goats.
The study is well written, except for some unclear sentences I highlighted in the line comments. It is a modest study, but authors honestly discussed the limits, which I appreciated. The approach using online survey is uncommon, but interesting, and allows to have first sights about a phenomenon understudied in the scientific literature and which is in the scope of wild/domestic interaction. In this sense, I think the paper is well suited for publication in the special issue.
I nevertheless have some comments and interrogations that need to be addressed. For instance, I think that introduction and discussion could be enlarged and included in a wider scientific framework, using more references and comparisons with other species. Results and tables also need clarification. Questions, comments and suggestions are reported in the general and line comments below.
General comments
- In the summary, authors mentioned that they aim to map the distribution of hybridization events in time and space. But no discussion about the evolution of hybrids locations with time is reported.
- In the introduction, a first paragraph explaining why hybridization and introgression are a threat for wild species, in a more general context would be of interest for the readership. For instance, by also linking this to the scope of the special issue the paper is submitted in: from Genotype to Phenotype in the context of wild/domestic interaction.
- Authors mention in the paper more than once that they also included data from scientific and grey literature, but there are no details in the main text that explain which data (e.g. how many hybrids reported) come from literature and how it was accounted for in the final map.
- The discussion lacks of comparison with other species, and biological systems. It also lacks some biological interpretations that would enlarge the scope of the paper and improve the discussion quality. I highlighted several points in line comments to suggest to the authors some topics to broach and also suggested some restructuring. One additional point to discuss could be about the fact that, despite a reduced reproductive success observed in hybrids (as said in introduction), free ranging hybrids seem to persist at a non-negligible rate (since authors said that 48 was higher than expected). Which mechanisms can explain that? On the opposite, 48 can be seen as a small number, that may be explained by the lower survival of hybrids and their lower fertility, avoiding hybridization to reach high levels.
- The approach using online surveys and relying on expert opinions and photographers or naturalists is original and illustrate how citizen science might help conservation. It is a point which is not discussed or highlighted in the paper but should be, since it is of interest.
Lines comments
L12-16: this sentence is too long. A suggestion of rephrasing can be: “The species occasionally suffers local population reductions as a consequence of epidemic diseases. While this phenomenon is well studied***, another threat for the species long-term conservation is hybridization with feral domestic goats, which is a lot less investigated.”
*** is it true??
L17: if I am not wrong, no time mapping has been done nor discussed. Authors reports hybrids from 2000 to 2021 but do not discuss the evolution of hybrids locations with time. It could be interesting though.
L19: Authors mention in the paper more than once that they also included data from scientific and grey literature, but there are no details in the main text that explain which data (e.g. how many hybrids reported) come from literature.
L20: “suggested” instead of “showed” since hybrids have not been genetically confirmed (except few of them).
L21: here and in the abstract, authors say “more hybrids than expected”. But in the article, nothing is detailed about how many are expected, or based on what we could expect a lower hybridization rate than evidenced here. This should be made clearer.
L22-25: I would rather say that this study calls for more studies about ibex hybridization, but I am not sure it could, by itself, serve as a baseline for conservation. A rephrasing suggestion is “This calls for more studies clearly quantifying hybrids in Alpine ibex colonies but also highlights the need for conservation actions aiming to control this phenomenon, such as the effective management of domestic herds grazing in Alpine ibex core areas and eventually the removal of hybrids. Our results also illustrated the urgent need of establishing a standardized protocol to morphologically identify hybrids in the field as well as a formal description of morphological traits characterizing hybrids”.
This last point is highlighted in the discussion and is, I think, one of the main conclusions of the study: identifying hybrids and managing hybridization is challenging. It should be reported in the abstract and summary.
L28: Although “being currently classified as”
L29: “a potential threat for its long-term conservation is introgression following hybridization with domestic goats”
L30: “Hybridization has been documented in Switzerland in captive and free ranging animals but accurate data to assess the extent of this phenomenon in natural condition on the Alps are lacking.”
L32-34: sentence too long. “Using an online survey and a network of experts, as well as information available in scientific and grey literature, we collected and mapped unpublished evidence of hybridization events that occurred between Alpine ibex and domestic goat from 2000 to 2021.”
L36: more common than expected based on? As previously, this point should be addressed in the introduction.
L36-40: same comment as L22-25
L54: “The strong bottleneck that occurred” or “which occurred” + delete the coma after “[2-4]”
L56-57: Not sure to understand the point of mentioning inbreeding depression here.
L57-59: same rephrasing as earlier
L59-61: need a reference to this statement about increasing hybridization due to human activities.
L72: typo, only one d at description
L105-106: what are these spatial units? the countries? or within countries, smaller spatial units like populations or massifs known to host Capra ibex?
L115-121: what about the data from the scientific literature and the grey literature? how was it included in the study? (as mentioned L95-97). Was the search for scientific literature exhaustive? What were the key words used? Have the authors used Web of Science?
L118: the url is just saying that the survey ended. Not sure of the utility of putting this url in the paper. But the list of questions might be interesting to report, at least in supplementary material. I saw by the end of my reviewing process that the form was available for me to download with the manuscript. This supplementary material should be called in the main article.
L160: what is considered as a colony? It is specified L164 that they are defined as in ref 21, but I think this should be made explicit here. A bit more details would be appreciated
L160: “from the same area”. Which spatial scale is this area? the locality? the municipality? was it based on the vital domain size of Capra ibex? Or is it the colony?
L165: and if coordinates were not available, how were considered the reports?
Table 1: In Germany, ref 21 reports 5 colonies, while here 4 have been investigated, and the % of colonies covered is 100%. It should be 4/5 = 80%. Similarly, for France, 6/30 = 20%. Can the author explain? The number of total colonies cited in the text is 178, as in ref 21. So maybe it is because several reports for a same colony have been collected? (for France at least, because for Germany this doesn't explain the value of 100%). Clarification is needed.
L188: from where is this total? I guess ref 21, but it needs to be stated here
L199: 34+3=37: the last 11 individuals are Q3? We learn in the following that yes, but it should appear here also, so that the reader does not wonder.
L217: add “see ref 21”
Figure 1: I really appreciated this figure, it is nice to visualize the patterns. However, precise definitions of what are the characteristics that make the experts say "yes, for sure it’s an ibex" should be provided, at least in supplementary material. I thought the same about which morphological characteristics allow to say “it is a hybrid” but in the discussion this point is broach and authors explain that such formal description is lacking and is hard to produce because of phenotypic variability. This is an important point of the study, and it should be highlighted in the abstract (see previous comments), and emphasized in the discussion.
Figure 2: indicate that the total number of colonies and their locations are from ref 21
L241-257: this first paragraph of discussion should be in introduction: the difference in number of reports between peer-reviewed and grey papers justifies the present study and explain in what such an approach is needed
L244: “peer-reviewed papers” instead of “peer review papers”
L246-247: don't understand what is the meaning of this sentence is. Is this paper the only one published after Iacolina et al. 2019, and does it report about Alpine ibex?
L247-249: This should be in results if literature data was included in the map. If it was not, then rephrasing is needed at the beginning of the paper because it let the reader think that literature data is accounted for.
L260: delete coma
L261: I suggest rephrasing for “from most of the Alpine ibex geographic range”
L266-267: I don't understand these values. Table 1 reports 2 reports for Austria, with one hybrid. Then at maximum 2 colonies were investigated. If one hybrid is reported in one of two colonies, and the % here are the % of colonies in which hybrids have been counted among the investigated colonies (as I understand from "20.9% of the investigated colonies" L266), then it should be 50% for Austria.
L265-271: this is almost results, unless authors provide some biological interpretation: what can explain a higher rate of hybridization in Italy than in other countries? does this rate of hybridization corresponds to what can be observed in other species? more particularly, ungulate species for which domestic conspecifics exist?
I guess some explanation comes from information provided L313-329. The discussion might be better structured to discuss the different results.
L271: does the rate of feral goats correlates with the rate of hybrids? similar questions for the number of goat shepherds or herds in ibex areas? Information given in L296-312 should appear earlier in the discussion
L279-282: this sentence is unclear and too long. What is "the distance of the hybrid phenotype to the phenotype of the parental species with some individuals looking more similar to domestic goats and other looking more similar to the Alpine ibex"?
L279-295: this paragraph should be, for me, an ending paragraph of the discussion. The next one is really interesting discussion and should appear earlier in the discussion because it answers question readers ask themselves since a long time while reading the paper.
L293: highlighted the need of establishing a reliable protocol to identify hybrids and describe more precisely phenotypic variation of Alpine ibex.
L304-306: how was density accounted for? it is unclear to which result we are referring here
L342: “with what can be expected” instead of “with what expected”. But I would rephrase for something like "the fact that suspected hybrids were more abundant in the areas with higher density of Alpine ibex (i.e., Western Alps) comforts the reliability of our survey approach". Since we could "expect" a high density of ibex but no goat and thus no hybrids.
L361: it seems that there is more hybridization occurs than previously known by the authors (see might previous comments about the need of justification for this expectation). But 48 among all the > 50 000 individuals reported in the alps (ref 21) seem to me a low number. Authors should make clearer why they state that they reveal a "high rate". I am not an expert of ibex so maybe I don't have the background to understand, but other readers will maybe be in the same case.
Here "neglected" maybe refers to that fact that only few genetic studies focused on hybridization, and I agree this is a problem. But this point should be made clearer if it is the meaning of this sentence.
L365: Typo here: “Rossi et al. 2019” is redundant with “[11]”
L368: levels (plural)
Author Response
Reviewer 3
In this paper, authors investigated through an online survey and a literature review, the occurrence of Alpine ibex x Capra hircus hybrids in the Alps. Investigating almost half of the total number of colonies known in the species range, they report the presence of 48 suspected hybrids. They conclude that their study reveals a higher than expected number of hybrids and highlight the need of a better management policy of hybrids and feral goats.
The study is well written, except for some unclear sentences I highlighted in the line comments. It is a modest study, but authors honestly discussed the limits, which I appreciated. The approach using online survey is uncommon, but interesting, and allows to have first sights about a phenomenon understudied in the scientific literature and which is in the scope of wild/domestic interaction. In this sense, I think the paper is well suited for publication in the special issue.
I nevertheless have some comments and interrogations that need to be addressed. For instance, I think that introduction and discussion could be enlarged and included in a wider scientific framework, using more references and comparisons with other species. Results and tables also need clarification. Questions, comments and suggestions are reported in the general and line comments below.
Thank you very much for your constructive comments, we have edited the text accordingly and you can find our replies below each comment.
General comments
- In the summary, authors mentioned that they aim to map the distribution of hybridization events in time and space. But no discussion about the evolution of hybrids locations with time is reported.
Response: we have removed the reference to time as no temporal analysis were provided.
- In the introduction, a first paragraph explaining why hybridization and introgression are a threat for wild species, in a more general context would be of interest for the readership. For instance, by also linking this to the scope of the special issue the paper is submitted in: from Genotype to Phenotype in the context of wild/domestic interaction.
Response: we have moved and extended the paragraph about anthropogenic hybridization at the beginning of the introduction, and we have widened the context of our study by adding references to other species.
- Authors mention in the paper more than once that they also included data from scientific and grey literature, but there are no details in the main text that explain which data (e.g. how many hybrids reported) come from literature and how it was accounted for in the final map.
Response: data from scientific and grey literature were used mostly before the study to try to quantify the phenomenon. All the data used for the research came instead from the survey and from direct reports. We have edited the text in order to avoid this misunderstanding.
- The discussion lacks of comparison with other species, and biological systems. It also lacks some biological interpretations that would enlarge the scope of the paper and improve the discussion quality. I highlighted several points in line comments to suggest to the authors some topics to broach and also suggested some restructuring. One additional point to discuss could be about the fact that, despite a reduced reproductive success observed in hybrids (as said in introduction), free ranging hybrids seem to persist at a non-negligible rate (since authors said that 48 was higher than expected). Which mechanisms can explain that? On the opposite, 48 can be seen as a small number, that may be explained by the lower survival of hybrids and their lower fertility, avoiding hybridization to reach high levels.
We have widely restructured the discussion. We have better explained the differences between what we were expecting and what we found and also added more biological interpretation of our results as well as discussed possible causes and consequences of it.
- The approach using online surveys and relying on expert opinions and photographers or naturalists is original and illustrate how citizen science might help conservation. It is a point which is not discussed or highlighted in the paper but should be, since it is of interest.
We agree that this point was missing in the discussion, and we have therefore added it.
Lines comments
L12-16: this sentence is too long. A suggestion of rephrasing can be: “The species occasionally suffers local population reductions as a consequence of epidemic diseases. While this phenomenon is well studied***, another threat for the species long-term conservation is hybridization with feral domestic goats, which is a lot less investigated.” *** is it true??
Response: the sentence has been modified to be more clear.
L17: if I am not wrong, no time mapping has been done nor discussed. Authors reports hybrids from 2000 to 2021 but do not discuss the evolution of hybrids locations with time. It could be interesting though.
Response: Yes, you are right. The data collected did not allow to perform any temporal analysis so we have deleted the reference to it.
L19: Authors mention in the paper more than once that they also included data from scientific and grey literature, but there are no details in the main text that explain which data (e.g. how many hybrids reported) come from literature.
Response: we agree with you that this should be deleted from the actual “scope” of the study. We did not include reports from the scientific literature in the data collected from our survey, although we mention and discuss the previous reports of hybridization between Alpine ibex and feral domestic goats in Introduction section and discussion pointing out the limited available scientific literature on this topic. Thus, we deleted “as well as information available in scientific literature” in the text.
L20: “suggested” instead of “showed” since hybrids have not been genetically confirmed (except few of them).
Response: modified as suggested.
L21: here and in the abstract, authors say “more hybrids than expected”. But in the article, nothing is detailed about how many are expected, or based on what we could expect a lower hybridization rate than evidenced here. This should be made clearer.
Response: this point was raised as well by another reviewer. With “hybrids more common than expected” we meant that, according to the sparse scientific literature on this topic, hybrids were expected to be only a limited, sporadic phenomenon. On the contrary, anecdotical reports pointed towards a non-rare occurrence of the phenomenon. Our study proved that the latter hypothesis was correct, as not only many phenotypic hybrids have been reported in the Alps, but also in our results there might be an underestimation, as stated in Discussion “Accordingly, we can’t exclude that hybrids or feral goat were present in more zones than the ones reported here.”. We acknowledge that the sentence was not clear and that there was some incoherence in the text. Therefore, we have modified the sentence to gain clarity.
L22-25: I would rather say that this study calls for more studies about ibex hybridization, but I am not sure it could, by itself, serve as a baseline for conservation. A rephrasing suggestion is “This calls for more studies clearly quantifying hybrids in Alpine ibex colonies but also highlights the need for conservation actions aiming to control this phenomenon, such as the effective management of domestic herds grazing in Alpine ibex core areas and eventually the removal of hybrids. Our results also illustrated the urgent need of establishing a standardized protocol to morphologically identify hybrids in the field as well as a formal description of morphological traits characterizing hybrids”.
This last point is highlighted in the discussion and is, I think, one of the main conclusions of the study: identifying hybrids and managing hybridization is challenging. It should be reported in the abstract and summary.
Response: thank you for the suggestion, we modified the paragraph as suggested: “”
L28: Although “being currently classified as”
Response: modified as suggested.
L29: “a potential threat for its long-term conservation is introgression following hybridization with domestic goats”
Response: modified as suggested.
L30: “Hybridization has been documented in Switzerland in captive and free ranging animals but accurate data to assess the extent of this phenomenon in natural condition on the Alps are lacking.”
Response: modified as suggested.
L32-34: sentence too long. “Using an online survey and a network of experts, as well as information available in scientific and grey literature, we collected and mapped unpublished evidence of hybridization events that occurred between Alpine ibex and domestic goat from 2000 to 2021.”
Response: modified as suggested.
L36: more common than expected based on? As previously, this point should be addressed in the introduction.
Response: see answer above, modification has been done to gain clarity.:
L36-40: same comment as L22-25
Response: modified as suggested..
L54: “The strong bottleneck that occurred” or “which occurred” + delete the coma after “[2-4]”
Response: modified as suggested.
L56-57: Not sure to understand the point of mentioning inbreeding depression here.
Response: We have edited the sentence giving a more general and less detailed overview of the genetic issues that threaten Alpine ibex long-term conservation.
L57-59: same rephrasing as earlier
Response: rephrased in the text as suggested
L59-61: need a reference to this statement about increasing hybridization due to human activities.
Response: we added as a reference Iacolina et al (2019)
L72: typo, only one d at description
Response: modified as suggested.
L105-106: what are these spatial units? the countries? or within countries, smaller spatial units like populations or massifs known to host Capra ibex?
As we don’t discuss the results based on those spatial units, we have removed the reference to them and only left the reference to the countries of presence of the species in the wild.
L115-121: what about the data from the scientific literature and the grey literature? how was it included in the study? (as mentioned L95-97). Was the search for scientific literature exhaustive? What were the key words used? Have the authors used Web of Science?
Response: As mentioned in the answer above, we deleted “as well as information available in scientific literature” in the introduction, so that now, this part should not appear in methods neither.
L118: the url is just saying that the survey ended. Not sure of the utility of putting this url in the paper. But the list of questions might be interesting to report, at least in supplementary material. I saw by the end of my reviewing process that the form was available for me to download with the manuscript. This supplementary material should be called in the main article.
Response: Thank you for pointing out this issue. We have delete the url in the text, adding the questionnaire (English version) in the form of a PDF file in the supplementary materials.
L160: what is considered as a colony? It is specified L164 that they are defined as in ref 21, but I think this should be made explicit here. A bit more details would be appreciated
Response: this issue has been raised by reviewer 4 as well. We added the definition of the colony in the first paragraph of the 2.3 section which now also include a more clear description of the data collected and on the different datasets.
L160: “from the same area”. Which spatial scale is this area? the locality? the municipality? was it based on the vital domain size of Capra ibex? Or is it the colony?
Response: we considered reports from the “same area” those that were reported in the same valley or municipality. We clarified it in the text as well.
L165: and if coordinates were not available, how were considered the reports?
Response: If coordinates were missing, the reports were considered at the municipality level (which was always reported), assigning then the observation to the associated ibex colony.
Table 1: In Germany, ref 21 reports 5 colonies, while here 4 have been investigated, and the % of colonies covered is 100%. It should be 4/5 = 80%. Similarly, for France, 6/30 = 20%. Can the author explain? The number of total colonies cited in the text is 178, as in ref 21. So maybe it is because several reports for a same colony have been collected? (for France at least, because for Germany this doesn't explain the value of 100%). Clarification is needed.
Response: The number of reports does not necessarily match with the number of colonies. This means that the equation “N reports= N colonies” should not be applied. This is because from a single person/authority we may have received reports covering more than a single ibex colony, as it happened for German colonies, where we received only negative reports and one report was about two different neighboring colonies. On the other side, we also received several reports from different people covering the same colony, this is why we have 124 reports from Italy but only 62% of colonies covered. We have added the explanation also in the caption of the table to avoid misunderstandings.
L188: from where is this total? I guess ref 21, but it needs to be stated here
Response: yes, the total number of Alpine ibex colonies per country is defined as in ref 21. We have made this explicit in the caption and also added the total number of colonies covered and the total number of colonies per country in Table 1 for the sake of clarity.
L199: 34+3=37: the last 11 individuals are Q3? We learn in the following that yes, but it should appear here also, so that the reader does not wonder.
Response: Yes, Q3= 11 reports. This information is reported in line Table 2. To avoid misunderstanding, we have also added the total number of Q1 Q2 and Q3 reports immediately after the total number of reports (Table 2, line 332).
L217: add “see ref 21”
Response: We have added the reference as suggested.
Figure 1: I really appreciated this figure, it is nice to visualize the patterns. However, precise definitions of what are the characteristics that make the experts say "yes, for sure it’s an ibex" should be provided, at least in supplementary material. I thought the same about which morphological characteristics allow to say “it is a hybrid” but in the discussion this point is broach and authors explain that such formal description is lacking and is hard to produce because of phenotypic variability. This is an important point of the study, and it should be highlighted in the abstract (see previous comments) and emphasized in the discussion.
Response: We highlighted this point in the abstract as suggested by the reviewer (see response above). Moreover, in the results section we highlighted that “The phenotype of the individuals identified as probable or confirmed hybrids was highly variable and included variations in the size and shape of the horns, of the body outline and of the coat color compared to Alpine ibex. Some individuals were also hornless”.
The main scope of the expert evaluation (+ the classification in reliability categories) was to give the most reliable visual evaluation in the absence of clear-cut guidelines on how to morphologically recognize hybrids Alpine ibex x domestic goat. Therefore, we cannot provide a precise definition on specific characteristics that describe the phenotype of hybrids, as this was out of the scope of this study. However, as we realized that this point is crucial, we have deepened it also in the discussion.
Figure 2: indicate that the total number of colonies and their locations are from ref 21
Response: we have rewritten the figure caption to better explain the map and we have also added the reference.
L241-257: this first paragraph of discussion should be in introduction: the difference in number of reports between peer-reviewed and grey papers justifies the present study and explain in what such an approach is needed
Response: we moved the entire paragraph in Introduction section
L244: “peer-reviewed papers” instead of “peer review papers”
Response: modified as suggested.
L246-247: don't understand what is the meaning of this sentence is. Is this paper the only one published after Iacolina et al. 2019, and does it report about Alpine ibex?
Response: The sentence means that a recent paper (2022) on the hybridization phenomenon of Alpine ibex was published after the review of Iacolina. To our knowledge, this paper is the only one published after the review of Iacolina. To improve clarity, we modified the sentence in the text
L247-249: This should be in results if literature data was included in the map. If it was not, then rephrasing is needed at the beginning of the paper because it let the reader think that literature data is accounted for.
Response: this point has been clarified in the previous replies. Results of this study includes only reports from the survey.
L260: delete coma
Response: modified as suggested.
L261: I suggest rephrasing for “from most of the Alpine ibex geographic range”
Response: modified as suggested.
L266-267: I don't understand these values. Table 1 reports 2 reports for Austria, with one hybrid. Then at maximum 2 colonies were investigated. If one hybrid is reported in one of two colonies, and the % here are the % of colonies in which hybrids have been counted among the investigated colonies (as I understand from "20.9% of the investigated colonies" L266), then it should be 50% for Austria.
Response: Thank you for pointing this out. There was a mistake in the text. Actually, the % reported here are the % of colonies in which hybrids have been counted among the total number of colonies. In Austria there are 27 colonies (as defined in Brambilla et al., 2020), thus 1/27= 3,7%. As suggested by reviewer 4, however, this numbers have been moved to the results section. We have also added a sentence to warn about possible bias of these %.
L265-271: this is almost results, unless authors provide some biological interpretation: what can explain a higher rate of hybridization in Italy than in other countries? does this rate of hybridization corresponds to what can be observed in other species? more particularly, ungulate species for which domestic conspecifics exist?
I guess some explanation comes from information provided L313-329. The discussion might be better structured to discuss the different results.
Response: Yes, indeed one of the possible explanations of the clusters reported in Italy was explained in the Discussion.. We moved the sentence so that now the paragraphs are more connected.
L271: does the rate of feral goats correlates with the rate of hybrids? similar questions for the number of goat shepherds or herds in ibex areas? Information given in L296-312 should appear earlier in the discussion
Response:The data about feral domestic goats only included the information about their presence in the area, without quantification of the number of individuals (usually one but may be variable). Therefore, we could not provide any analysis about this. Similarly, we don’t have detailed data on the number of herds/shepherd in each of the colonies, hence, no numerical analysis were done. We have however added this issue in the discussion.
L279-282: this sentence is unclear and too long. What is "the distance of the hybrid phenotype to the phenotype of the parental species with some individuals looking more similar to domestic goats and other looking more similar to the Alpine ibex"?
Response: the sentence has been rephrased as follows (lines 574-577) “While some individuals looked more similar to domestic goats, others were phenotypically more similar to Alpine ibex, as the genetic distance between parental species (which may have been pure Alpine ibex in some cases, or first/second generation back-crosses in others) and domestic goats was variable.”
L279-295: this paragraph should be, for me, an ending paragraph of the discussion. The next one is really interesting discussion and should appear earlier in the discussion because it answers question readers ask themselves since a long time while reading the paper.
Response: The whole discussion has been restructured and this topic, further discussed, is now presented at the end of the section.
L293: highlighted the need of establishing a reliable protocol to identify hybrids and describe more precisely phenotypic variation of Alpine ibex.
Response: we have restructured the discussion as suggested in other comments. The concept of this sentence has been added and discussed extensively as suggested.
L304-306: how was density accounted for? it is unclear to which result we are referring here
Response: we changed “density” with “abundance”
L342: “with what can be expected” instead of “with what expected”. But I would rephrase for something like "the fact that suspected hybrids were more abundant in the areas with higher density of Alpine ibex (i.e., Western Alps) comforts the reliability of our survey approach". Since we could "expect" a high density of ibex but no goat and thus no hybrids.
Response: modified as suggested.
L361: it seems that there is more hybridization occurs than previously known by the authors (see might previous comments about the need of justification for this expectation). But 48 among all the > 50 000 individuals reported in the alps (ref 21) seem to me a low number. Authors should make clearer why they state that they reveal a "high rate". I am not an expert of ibex so maybe I don't have the background to understand, but other readers will maybe be in the same case.
Here "neglected" maybe refers to that fact that only few genetic studies focused on hybridization, and I agree this is a problem. But this point should be made clearer if it is the meaning of this sentence.
Response: Hybridization in Alpine ibex is a relatively recent investigated phenomenon. We believe that hybridization is a neglected phenomenon not only because of the few scientific report/genetic studies present in the literature, but also because the authorities are not taking appropriate and homogenous actions to tackle this problem. Moreover, considering that Alpine ibex is a species with a low genetic diversity (as explained in Introduction section) and of conservation concern, the report of 48 free-roaming hybrids that might continue to breed with ibex for several generations if not contained, can be worrying. We have however discussed all the possible implication of our observation and we now believe that the discussion is more equilibrate.
L365: Typo here: “Rossi et al. 2019” is redundant with “[11]”
Response: deleted as suggested.
L368: levels (plural)
Response: modified as suggested.

Reviewer 4 Report
Dear editor/authors,
Manuscript of Moroni et al. „ Hybridization between Alpine ibex and domestic goat in the Alps: a sporadic and localized phenomenon?“ submitted for evaluation to the Animals as a research paper presents a very interesting set of data. The topic is very current and the manuscript is very well written.
General comments:
Several terms run through the text for feral/abandoned goats and domestic goats. This problem is well known, so these terms should be well distinguished because they are confused on several occasions through the MS. In my opinion, feral/abandoned goats (without owner) are not under human supervision and control at any time, i.e. they roam freely in mountains all year round. Domestic goats (have an owner) are under human control, they can stay part of the year in the mountains and create a problem with ibex, but they can be controlled and descended from the mountain in time (i.e. close monitoring of the livestock populations and herding practices to be constantly incorporated into wildlife conservation; managers should establish buffer zones between livestock). The question remains what to do with a feral/abandoned animal, e.g. in my country we are allowed to hunt them after obtaining special permits from state institutions.
I am aware that it is not in your MS focus, but I think that the appearance of hybrids is significantly related to the number of livestock in the mountains. So, I wonder if you can provide some official number of livestock for areas where hybridization occurred (some basic statistical analysis).
Specific comments:
Keywords are repeated as in the title, so change the following: Alpine ibex to Capra ibex; domestic goat to Capra hircus; hybridisation to crossbreeding.
L20 and L35: but in the results, you mention one case in Austria (Table1; L210-212) why it is not here?
L72: ddescription; one letter “d” is redundant
The last sentence from the study area “In parallel to the distribution…” belongs to 2.2. Data collection paragraph
Figure 1: please can you add information on where it was photographed
Figure 2: as a resident of Europe, I know which part it is but to other people it will be very difficult. So, I suggest you indicate the countries and insert a small map of Europe where the location of your detailed map will be seen. For geographic distribution of ibex, do you use the IUCN Red List of Threatened Species data? If yes, then state it in the text. Sorry, but the legend with the colors is not the clearest to me, please give a better explanation!
Author Response
Reviewer 2
Dear editor/authors,
Manuscript of Moroni et al. „ Hybridization between Alpine ibex and domestic goat in the Alps: a sporadic and localized phenomenon?“ submitted for evaluation to the Animals as a research paper presents a very interesting set of data. The topic is very current and the manuscript is very well written.
General comments:
Several terms run through the text for feral/abandoned goats and domestic goats. This problem is well known, so these terms should be well distinguished because they are confused on several occasions through the MS. In my opinion, feral/abandoned goats (without owner) are not under human supervision and control at any time, i.e. they roam freely in mountains all year round. Domestic goats (have an owner) are under human control, they can stay part of the year in the mountains and create a problem with ibex, but they can be controlled and descended from the mountain in time (i.e. close monitoring of the livestock populations and herding practices to be constantly incorporated into wildlife conservation; managers should establish buffer zones between livestock). The question remains what to do with a feral/abandoned animal, e.g. in my country we are allowed to hunt them after obtaining special permits from state institutions.
Response: Thank you for pointing out this terminology confusion. We use domestic goat as the common name of the species Capra hircus, without meaning that they have an owner. To reduce possible misunderstanding, we have decided to stick to the term “feral domestic goats” when making explicit reference to the way these animals are managed (i.e., don’t have an owner) while we left “domestic goat” when we were referring to the species in general.
I am aware that it is not in your MS focus, but I think that the appearance of hybrids is significantly related to the number of livestock in the mountains. So, I wonder if you can provide some official number of livestock for areas where hybridization occurred (some basic statistical analysis).
Response: We agree with you that hybrid appearance is significantly related to Alpine livestock grazed near Alpine ibex colonies. Comparing our results (hybrid reports) with domestic caprine near Alpine ibex colonies would be quite interesting, although it would require official zootechnic data in repository from different countries, for which we don’t have access. We added the following sentence in Discussion section (lines 465-470):
“It is likely that the presence of feral domestic goats is correlated with the number of domestic goats and goat herds that pasture in the mountains during summer in a given area. Unfortunately, while official data at the national or infranational scale point towards much greater goat numbers in the Western than in the Eastern Alps, with a ratio well above 2:1, this data was not available at a smaller scale (a more relevant one for the scope of this research), and therefore we could not test this hypothesis.”
Specific comments:
Keywords are repeated as in the title, so change the following: Alpine ibex to Capra ibex; domestic goat to Capra hircus; hybridisation to crossbreeding.
Response: thank you for this suggestion. We modified as suggested.
L20 and L35: but in the results, you mention one case in Austria (Table1; L210-212) why it is not here?
Response: to avoid confusion, we have removed all references to the countries from the abstract and then made explicit reference to them in the text (lines…)
L72: ddescription; one letter “d” is redundant
Response: corrected as suggested.
The last sentence from the study area “In parallel to the distribution…” belongs to 2.2. Data collection paragraph
Response: we changed the position of the sentence to line 205-207, in 2.2
Figure 1: please can you add information on where it was photographed
Response: We added information on the location of the photographs.
Figure 2: as a resident of Europe, I know which part it is but to other people it will be very difficult. So, I suggest you indicate the countries and insert a small map of Europe where the location of your detailed map will be seen. For geographic distribution of ibex, do you use the IUCN Red List of Threatened Species data? If yes, then state it in the text. Sorry, but the legend with the colors is not the clearest to me, please give a better explanation!
Response: The map (Figure 2) was amended as suggested. A focus on Europe and names of the countries were added. In addition, we gave a better explanation of the legend in the figure caption, as follows:
“Distribution and abundance of the probable hybrids observed between 2000 and 2021 in the Alps and localization of the feral goats observed in the same areas. The map includes the colonies of Alpine ibex (data originally from [33], also included in the IUCN Red List of Threatened Species [12]) coloured according to the number of hybrids observed. The orange to dark-red gradient represents colonies with number of hybrids reported between 1 and 20 (with reports of quality Q1 and Q2). Yellow polygons represent colonies for which a single report of probable hybrid was received but the quality of the report was low (Q3: report coming from reliable local experts but with no or low-quality visual documentation available). White polygons represent colonies where no hybrids were reported (negative reports) while dark grey polygons represent colonies for which no reports were available (neither of presence nor of absence of hybrids). Alpine ibex colonies where also feral goats were observed are represented with a blue dot.”

Round 2
Reviewer 2 Report
Comments to the Author
Overall, I think the manuscript is much improved and I am satisfied with the revisions and responses to comments. This is a nice paper. I have just a few very minor suggestions:
L. 48: I do not understand well this sentence. I think one word is missing, or on the contrary, extra.
L. 53 : « known » instead of « know »
L. 57: I would add a comma after “Canis lupus” and change “for which” instead of “which” or replace by “where”.
L. 308: Should not it be “feral domestic goats” to be consistent?
L. 335: Should not it be “feral domestic goats” to be consistent?
L. 387-393: This sentence is a bit long. What do you think about cutting it?
Author Response
Reviewer 2
Overall, I think the manuscript is much improved and I am satisfied with the revisions and responses to comments. This is a nice paper. I have just a few very minor suggestions:
- 48: I do not understand well this sentence. I think one word is missing, or on the contrary, extra.
Response: We deleted “between” in line 48
- 53 : « known » instead of « know »
Response: corrected as suggested
- 57: I would add a comma after “Canis lupus” and change “for which” instead of “which” or replace by “where”.
Response: modified as suggested
- 308: Should not it be “feral domestic goats” to be consistent?
Response: corrected as suggested
- 335: Should not it be “feral domestic goats” to be consistent?
Response: corrected as suggested
- 387-393: This sentence is a bit long. What do you think about cutting it?
Response: The sentence has been cut in two sentences, as follows (line 383):
The extensive genetic monitoring carried out in the last two decades in large part of the Alpine ibex range (through random sampling of phenotypically non-suspected individuals) [22,23,31,34] did not find evidence of ongoing hybridization [22]. This could suggest that, if non-adaptive or deleterious [35], domestic goat genes are quickly removed from the Alpine ibex populations (either because of possible reduced fecundity of hybrids [26] or through selection, or dilution, especially if a single hybridization event happens). "
